# Uncovering the Genetic Structure of the Sekler Population in Transylvania Through Genome-Wide Autosomal Data

**DOI:** 10.3390/genes17010030

**Published:** 2025-12-29

**Authors:** András Szabó, Zsolt Bánfai, Katalin Sümegi, Valerián Ádám, Ferenc Gallyas, Miklós Kásler, Béla Melegh

**Affiliations:** 1Department of Medical Genetics, Medical School, Clinical Centre, University of Pécs, Szigeti út 12., 7624 Pécs, Hungary; szabo.andras@pte.hu (A.S.); sumegi.katalin@pte.hu (K.S.); melegh.bela@pte.hu (B.M.); 2Department of Biochemistry and Medical Chemistry, Medical School, University of Pécs, Szigeti út 12., 7624 Pécs, Hungary; 3Institute of Hungarian Research, Andrássy út 64., 1062 Budapest, Hungary

**Keywords:** population genetics, ethnic minorities, Sekler, genome-wide autosomal data, SNP array, allele frequency, haplotype analysis

## Abstract

Background/Objectives: The Seklers are a Hungarian-speaking regional population in Transylvania, Romania, with a long and complex history, yet comprehensive genome-wide studies remain limited. Our aim was to characterize the genetic background of multiple Sekler communities using high-density autosomal data and to place them in a broader Central and Eastern European context. Methods: Here we analyzed genome-wide autosomal SNP data obtained from 17 Sekler groups. Allele frequency- and haplotype-based approaches were applied to assess overall genetic structure, ancestry patterns, recent shared ancestry, and signals of demographic history. Results: Analyses based on overall allele-frequency patterns showed that Sekler groups fit into a single, coherent genetic cluster shared with Hungarians. No major differences were detected among the Sekler communities at this broader genomic level, and their genetic profiles were largely indistinguishable from one another. Using haplotype-based methods, most Sekler groups again formed a compact cluster. However, two villages, Deményháza and Nyárádszentimre, showed clear signs of increased within-group relatedness and subtle separation. These patterns might indicate that both communities experienced stronger local drift and reduced effective population size, while other Sekler groups showed no comparable deviation from the general regional pattern. Conclusions: Although a small number of villages display modest signs of localized demographic drift, our results support that the Seklers represent a regionally distinct and internally cohesive population, whose genetic structure is shaped mainly by common historical and linguistic ties, with minor village-level variation, forming a uniform part of the Hungarian-speaking population of the East-Central European region.

## 1. Introduction

The Carpathian Basin has a complex population history shaped by multiple prehistoric and historical migration events. Archaeogenomic studies show that the region experienced several major demographic shifts, beginning with the spread of Neolithic farmers from Anatolia (ca. 6000–4500 BCE), followed by substantial Bronze Age gene flow associated with Yamnaya-related steppe groups (ca. 3000–2500 BCE) [1,2,3,4]. Later periods added further layers through Iron Age Scythian and Celtic movements, Roman-era population changes, and early medieval migrations involving Huns, Avars, and the ancestors of the Hungarians [5,6,7]. As a result, present-day Central and Eastern European populations generally carry mixed ancestry components derived from Western Hunter-Gatherers, Early European Farmers, and Bronze Age Steppe groups [2,3].

Although the population history of Hungarian-speaking groups has been investigated from historical, archaeological, and linguistic perspectives, high-resolution genomic data on specific regional subgroups remain limited. One such subgroup is the Seklers (Székelys), a Hungarian-speaking population living mainly in eastern Transylvania—present-day Harghita, Covasna, and parts of Mureș counties [8]. They are known for a strong cultural identity and for their historical role in defending the eastern borders of the medieval Kingdom of Hungary [8,9]. Their semi-autonomous administrative units (“Székely seats”) existed for centuries and contributed to the perception that the Seklers form a distinct community within the broader Hungarian population [9,10].

The origins of the Seklers have long been debated. Proposed explanations include descent from early Hungarian tribal groups, assimilation of earlier steppe-related populations such as Avars or Turkic groups, or historical connections to neighboring frontier communities including the Csángós [8,10]. Although older narratives sometimes suggested links to Huns or early Turkic peoples, current historical and genetic evidence does not support direct continuity [6,10].

Recent archaeogenomic studies of the Hungarian Conquest Period show that the conquering elite had highly mixed ancestry, including both European and Asian-related components [7,11]. Modern Hungarians, however, retain only a small fraction of these eastern signals (~2–4%), indicating extensive admixture with local European populations over the past millennium [12]. It remains unclear whether the Seklers differ from this general pattern, as genome-wide studies focusing specifically on them are scarce.

Available mtDNA and Y-chromosome data suggest that Seklers carry mostly West Eurasian haplogroups common in Central Europe, with only low levels of East Eurasian lineages [6,13]. Our earlier investigations of the Seklers, conducted on smaller and less representative datasets, have already pointed to a close genetic relationship with Hungarians from modern-day Hungary. Ancestral components typically show a mixture dominated by North–Central European ancestry, with some Balkan-related contributions and minor Asian-derived components [6,13,14,15]. These patterns reflect the long and complex demographic history of the region.

High-density autosomal SNP analyses allow finer resolution than uniparental markers because they capture genome-wide ancestry and can detect subtle population structure, drift, or founder effects [16,17]. In Central and Eastern Europe, such datasets have been particularly useful for distinguishing closely related groups, identifying isolation patterns, and assessing the impact of medieval and early modern demographic processes.

Given the historical significance of the Seklers and the long-standing debate about their origins and distinctiveness, a detailed genome-wide analysis provides an opportunity to place them into a broader regional context. Combining genetic data with archaeological, historical, and linguistic perspectives helps refine our understanding of how the Seklers fit into the wider population landscape of the Carpathian Basin.

The aim of this study is to use high-density autosomal SNP data to characterize the ancestry, population structure, and demographic history of Sekler communities. By comparing them with Hungarians, Romanians, and additional Eurasian reference populations, and by applying both allele frequency– and haplotype-based approaches, we seek to clarify their genetic position and identify any fine-scale structure or isolation patterns within the Sekler population.

## 2. Materials and Methods

### 2.1. Sample Groups and Genotype Analysis

Our dataset included 139 Hungarians from present-day Hungary and 456 Sekler individuals (15 from Magyarfenes, 64 from Tordaszentlászló, 13 from Székelykocsárd, 5 from Deményháza, 34 from Marosvásárhely, 22 from Nyárádköszvényes, 6 from Nyárádszentimre, 25 from Havad, 24 from Nyárádmente, 36 from Szováta, 23 from the region of Bukovina, 37 from Agyagfalva, 55 from Gyergyószentmiklós, 35 from Szentegyháza, 24 from Székelyudvarhely, 9 Unitarians from Korond and 29 from Kézdivásárhely) (Figure 1).

Ethnic background was self-reported by all participants. Recruitment occurred between 1 September 2022 and 31 March 2024. Eligibility required that participants have no recorded non-Hungarian-speaking ancestors within the past three generations, as verified through pedigree information.

All participants gave informed consent after verbal explanation of the study. The Regional Research Ethics Committee of Pécs approved the research protocol. All samples were anonymized, and the procedures adhered to the principles of the Declaration of Helsinki. Participants were recruited with the assistance of local clergy and general practitioners serving the respective settlements. Peripheral blood samples were collected by the local general practitioners into EDTA-anticoagulated tubes using standard clinical procedures.

Genomic DNA was extracted from peripheral blood leukocytes using standardized protocols. DNA extraction, quality control, and genome-wide genotyping were performed by the Human Genomics Facility (HuGe-F) (Human Genomics Facility, Erasmus MC, University Medical Center Rotterdam, Rotterdam, The Netherlands). DNA concentration and purity were assessed prior to genotyping using routine spectrophotometric and fluorometric methods to ensure suitability for downstream array-based analyses. Genome-wide genotyping was carried out using the Illumina Infinium Global Screening Array v3.0 BeadChip (Illumina Inc., San Diego, CA, USA), which contains approximately 650,000 SNPs, following the manufacturer’s recommended protocols. Data curation and quality control were conducted with Illumina GenomeStudio 2.0 (Illumina Inc., San Diego, CA, USA) and PLINK v1.9 (Broad Institute of MIT and Harvard, Cambridge, MA, USA) [18,19]. Hardy–Weinberg equilibrium (HWE) tests were applied separately to each population. A threshold of *p* < 1 × 10^−3^ was used for groups smaller than 50 individuals, while *p* < 1 × 10^−6^ was applied to larger cohorts, reflecting the likelihood that deviations from HWE may reflect demographic history rather than technical error. Variants with a minor allele frequency (MAF) < 0.05 were excluded, as were SNPs with > 10% missing calls. Samples with >5% missing genotypes would have been removed, although none met this condition. Genetic distances were assigned using the International Haplotype Map (HapMap) Phase 2 GRCh37 map [20]. Kinship analysis was performed to evaluate the degree of relatedness between individuals. Pairwise kinship coefficients were estimated using the KING 2.3.2 software (University of Virginia, Charlottesville, VA, USA) based on autosomal SNP genotypes. In order to avoid the confounding effects of close relatedness, for all pairs with a kinship coefficient >0.0884, one member of each pair was excluded from downstream analyses [21]. Following all filters, the dataset contained 595 individuals and 112,003 high-quality SNPs.

To place the investigated populations in a broader context, additional reference datasets were integrated. These included: (i) the Human Genome Diversity Project-Centre d’Étude du Polymorphisme Humain (HGDP-CEPH) European panel (*n* = 160, eight populations: Adygei, Basque, French, North Italian, Orcadian, Russian, Sardinian, Tuscan) genotyped on the Illumina 650Y array; (ii) the Romanian (*n* = 14), Latvian (*n* = 6), Caucasus (*n* = 158, nine populations: Abkhaz, Armenians, Balkar, Chechen, Georgian, Kumyk, Lezgin, Nogai, North Ossetian), Middle East (*n* = 118, eight populations: Kurdish, Turkish, Syrian, Jordanian, Iranian, Lebanese, Saudi, Yemeni) and Central Asian (*n* = 45, three populations: Turkmen, Uzbek, Tajik) samples obtained from the Estonian Biocentre repository, genotyped on various Illumina platforms; (iii) European populations from the 1000 Genomes Project (1 KGP, *n* = 264, including CEU, FIN, English only GBR) genotyped on the Illumina Infinium Omni2.5–8 array; and (iv) European populations from the HapMap Phase 3 Project (*n* = 253, including CEU and TSI) genotyped on Affymetrix and Illumina 1 M platforms [22,23,24,25,26,27,28,29]. Together, these public datasets provided comparative reference populations across Europe and Asia.

For clarity, the following abbreviations are defined here and are used consistently throughout the manuscript, including figures, tables and analyses. Population labels correspond to the following populations or geographic regions: SeklMagyarf—Magyarfenes; SeklTordaszl—Tordaszentlászló; SeklSzkocs—Székelykocsárd; SeklDemenyh—Deményháza; SeklMarosv—Marosvásárhely; SeklNyaradk—Nyárádköszvényes; SeklNyaradsz—Nyárádszentimre; SeklHavad—Havad; SeklNyarad—Nyárádmente; SeklSzov—Szováta; SeklBukov—Bukovina region; SeklAgyagf—Agyagfalva; SeklGyergy—Gyergyószentmiklós; SeklSzente—Szentegyháza; SeklSzudv—Székelyudvarhely; SeklUKor—Unitarians from Korond; SeklKezdi—Kézdivásárhely; GBRen—British (English only); CEU—Utah Residents with Northern and Western European ancestry; FIN—Finnish; TSI—Toscani in Italy.

### 2.2. Population Structure and Ancestry Analyses

To investigate genetic relationships among the Sekler populations and the considered Hungarian and publicly available reference groups described above, we applied three complementary methods.

Principal component analysis (PCA) was carried out using SMARTPCA, part of the EIGENSOFT v6.01 package (Broad Institute of MIT and Harvard, Cambridge, MA, USA) [30]. The statistical significance of each principal component was assessed with the built-in Tracy–Widom test, and the proportion of explained variance was also recorded. Pairwise fixation index (F_ST_) values were calculated with the same software to quantify average pairwise allele frequency differentiation between groups. The number of informative principal components was assessed using a scree plot. To identify the drop-off point of the calculated eigenvalues, we examined the curvature of the scree plot by computing the numerical second derivative of the plot. The drop-off point, corresponding to the maximum change in curvature (the elbow), occurred around PC6, after which the eigenvalues formed the characteristic flat tail. For downstream analyses, however, only the first four principal components were used.

Ancestral component inference was performed with ADMIXTURE v1.22 (University of California, Los Angeles, CA, USA), which estimates ancestry proportions under a maximum-likelihood clustering model [31]. The number of clusters was chosen based on cross-validation (CV) error minimization.

To explore deeper historical relationships, population splits and admixture events were modeled using TreeMix v1.13 (University of Chicago, Chicago, IL, USA), which reconstructs maximum-likelihood trees from genome-wide allele frequency data [32].

For these analyses, two main datasets were created. The first dataset contained Sekler and Hungarian samples, furthermore Romanian, Latvian, Caucasus, Middle East and Central Asian samples from the Estonian Biocentre, the HGDP-CEPH European samples and the European 1 KGP populations (CEU, FIN, English only GBR) for PCA and ADMIXTURE analyses. To minimize the confounding effects of linkage disequilibrium (LD), SNP pruning was applied using PLINK v1.9 with a sliding window of 50 SNPs, a step size of 5 SNPs, and an *r*^2^ threshold of 0.3. This slightly more permissive cutoff was chosen over the commonly used 0.2 to retain more informative markers while controlling LD effects [30,31]. After pruning, this dataset included 1360 individuals and 92,344 SNPs.

For clarity, PCA results were visualized both with and without the inclusion of Latvian, Caucasus, Middle East, Central Asian, HGDP-CEPH European and the European 1 KGP populations.

For TreeMix analyses, a second dataset was constructed including Sekler, Hungarian, Romanian, HGDP-CEPH European populations and Uyghur samples served as the outgroup. The pruned dataset comprised 779 individuals and 74,405 SNPs. A window size of 1000 SNPs was applied. Based on preliminary results and residual fit statistics, no migration edges were incorporated in the final model.

### 2.3. DNA Segment Analyses

Identity-by-descent (IBD) sharing was estimated genome-wide to quantify recent genetic relatedness among Seklers and reference populations [33]. Homozygosity-by-descent (HBD) was also assessed to evaluate the degree of autozygosity across populations. For IBD and HBD analysis, a third dataset was created incorporating Seklers and Hungarians, supplemented with Romanian, Latvian, Caucasus, Middle East and Central Asian samples from the Estonian Biocentre, the HGDP-CEPH European samples and HapMap Phase 3 CEU and TSI samples. For the IBD and HBD analyses, HapMap Phase 3 reference data were used instead of the 1000 Genomes Project dataset. The 1000 Genomes data are based on low-coverage whole-genome sequencing, which can result in uncertain genotype calls and reduced reliability in detecting identical-by-descent or homozygosity-by-descent segments. In contrast, the HapMap dataset provides high-quality, array-based genotypes with lower error rates, making it more suitable for these analyses. This unpruned dataset contained 1349 individuals and 111,301 SNPs.

Both IBD and HBD segments were detected using the Refined IBD algorithm implemented in Beagle v4.1 (University of Washington, Seattle, WA, USA). Alleles were recoded to the A2-major format in PLINK and subsequently converted into Variant Call Format (VCF v4.1) using PLINK/SEQ v0.10 (Broad Institute of MIT and Harvard, Cambridge, MA, USA) [34]. Parameters were set to a minimum IBD segment length of 3 cM, with trimming set to 10 and scaling adjusted according to √(n/100), where *n* is the number of samples (ensuring √(n/100) ≥ 2) [35]. All other parameters were left at default values. Average pairwise IBD sharing between populations was estimated using a standard population-level IBD calculation approach [36].

Average pairwise IBD-sharing values between each target population and all populations in the reference panel were computed from the RefinedIBD output. For each population, these values formed an “IBD profile” vector summarizing its genome-wide IBD connections to all other groups. To identify higher-order structure among populations based on their IBD profiles, we computed a Euclidean distance matrix between all population vectors. Classical multidimensional scaling (MDS) was then applied using the cmdscale function in R (R Foundation for Statistical Computing, Vienna, Austria), projecting the distance relationships onto two orthogonal axes (MDS1–MDS2) that best preserved the original inter-population distances [18,37]. The resulting coordinates were plotted with fixed population-wise color and symbol mappings to ensure visual consistency across PCA, HBD, and IBD-based analyses.

Because Refined IBD detects both shared and homozygous-by-descent tracts, it allowed us to infer levels of genomic autozygosity. For each individual, we computed both the average number and the cumulative length of HBD segments, enabling comparisons with populations previously characterized for isolation.

## 3. Results

### 3.1. Allele Frequency-Based Population Structure Analysis

PCA of the Sekler groups, Hungarians, Romanians, and several other Eurasian reference populations revealed the expected large-scale continental structure along PC1–PC2 and PC3–PC4 (Figure 2).

As shown on the upper PCA graphs, the Sekler samples cluster tightly with the Hungarian reference individuals and fall within the broader Central and Eastern European genetic continuum. None of the Sekler groups show any displacement toward the non-European reference clusters included in the PCA, such as the Caucasus, Middle Eastern, or Central Asian populations. Romanians form a partially overlapping but distinguishable cluster, consistent with their geographic and historical proximity. The bottom PCA graphs provide a focused view restricted to Seklers, Hungarians, and Romanians. Here, the Sekler groups form a compact cluster that overlaps almost entirely with the Hungarian samples, with no visible substructure separating individual Sekler villages from one another or from the Hungarian population. Romanians occupy an adjacent but slightly shifted position, as expected from known allele-frequency differences between the two populations. No Sekler subgroup shows an outlying or deviating position that would indicate distinct ancestry components or unique historical origins (Figure 2). Additional PCA results including further principal components are presented in Appendix A. These extended results confirm the same pattern: across all informative PCs, the Sekler individuals consistently cluster within the Hungarian genetic background, with no evidence for large-scale differentiation among Sekler communities.

The ADMIXTURE analyses (Figure 3; K = 3–5) were consistent with this overall pattern.

Across these K values, the Sekler groups exhibited the same ancestry components as the Hungarian reference population, with no group showing a distinct or unique signal suggestive of a separate origin. The full range of results (K = 2–10), provided in Appendix A, also showed no Sekler-specific component. Taken together, these findings support the view that the Sekler populations are genetically homogeneous when assessed through allele-frequency–based clustering.

The TreeMix analysis revealed a subtle but informative signal of internal differentiation. Although no migration edges were inferred, the tree showed slightly longer terminal branch for Nyárádszentimre and shorter for Deményháza—compared to other Seklers (Figure 4).

This suggests a somewhat different drift effect in these two villages. Importantly, this difference was not visible in PCA or ADMIXTURE, showing that TreeMix can pick up subtle drift even when allele frequency shifts remain small.

The F_ST_ analysis (Figure 5) also showed generally low differentiation within the Sekler dataset, with most pairwise values falling between 0.001 and 0.005, similar to internal variation seen among Hungarians.

The only notable exception was Nyárádszentimre, which showed consistently higher F_ST_ values (0.004–0.008) against all other Sekler populations, including the geographically closest ones such as the region of Nyárádmente and Nyárádköszvényes. Although these values are still low in absolute terms, they stand out clearly relative to the baseline Sekler–Sekler differentiation and match the drift pattern observed in TreeMix. In contrast, Deményháza did not show elevated F_ST_ and remained in the same range as other Sekler groups (0.001–0.003), despite its slightly less pronounced drift signature in haplotype-based analyses. This suggests that only drift in Nyárádszentimre has become strong enough to appear even in allele frequencies.

Overall, the allele frequency–based results show that the Sekler groups form a genetically homogeneous subset of the Hungarian-speaking population, with Nyárádszentimre being the only village that shows a measurable degree of divergence.

### 3.2. Identity-by-Descent Segment Analyses

The pairwise IBD-sharing matrix (Figure 6) showed a clear difference between most Sekler populations and two specific groups.

While the majority of Sekler communities had within-group IBD levels very similar to Hungarians, Deményháza and especially Nyárádszentimre showed a marked increase in within-group sharing. This means that individuals from these villages share noticeably more haplotypes with each other than individuals in other Sekler communities do.

Elevated IBD sharing typically reflects demographic factors such as small effective population size, endogamy, and limited historical gene flow. The high within-group values observed in Deményháza and Nyárádszentimre therefore strongly indicate that these two villages experienced a more isolated demographic history than the rest of the Sekler groups. In contrast, the other Sekler communities were again broadly similar to the Hungarian reference population, consistent with the allele frequency–based results.

To further explore the IBD patterns, we performed multidimensional scaling on the IBD distance matrix (Figure 7).

While the PCA gave a single compact Sekler cluster, the IBD-MDS revealed a very different structure. Most Sekler groups formed a coherent but loose cluster, with the exception of Deményháza and Nyárádszentimre, which two villages appeared as two clearly separated groups at distinct positions in the MDS space.

These separations were stable across different IBD thresholds and were visible along the main axes of variation. Since IBD-MDS captures recent shared ancestry, the results strongly suggest that these two villages have diverged demographically from the rest of the Sekler population over the last several generations. The close match between the IBD statistics and the MDS clustering provides independent confirmation that Deményháza and Nyárádszentimre form small, isolated subpopulations.

### 3.3. Autozygosity Analysis

The HBD results showed that most Sekler groups fall within a fairly narrow range of autozygosity, similar to the Hungarian reference population (Figure 8).

A few communities, such as Szentegyháza, Szováta and Agyagfalva, displayed slightly higher autozygosity, but these increases are not sufficient to indicate long-term or pronounced isolation on their own, and these groups did not stand out in any of the other analyses. In contrast, Deményháza and Nyárádszentimre showed somewhat elevated HBD levels that aligned with clear signals in the IBD and IBD-MDS analyses, where both villages consistently separated from all other Sekler groups. Positioned between the main Sekler cluster and the more pronounced outlier populations in the HBD plot, the slight increase observed in these two communities fits well with the broader haplotype-based evidence, suggesting that they represent smaller, more closed populations where genetic drift has acted more strongly over time.

## 4. Discussion

In this study, we examined several Sekler groups using both allele frequency-based and haplotype-based genomic analyses to assess their internal structure and their relationship to the Hungarians, Romanians, and other Eurasian populations. Overall, the allele frequency-based methods indicate that the Sekler groups form a coherent cluster within the broader genetic background of Hungarian-speaking populations, with only small deviations in a few villages. In contrast, haplotype-based approaches, which are more sensitive to recent shared ancestry and fine-scale demographic structure, reveal clear microstructural differences for two communities, Deményháza and Nyárádszentimre.

The allele frequency-based analyses are largely concordant. PCA places all Sekler samples in a single, tight cluster overlapping almost completely with Hungarians, and no Sekler subgroup separates as a distinct unit along the leading principal components. ADMIXTURE likewise reveals highly similar ancestry profiles across Sekler and Hungarian groups, even at higher K values, and no distinctive components emerge that would suggest a separate genetic origin for any Sekler subgroup. F_ST_ values between most Sekler groups are low and fall within the range expected for internal variation within a single population. One exception is Nyárádszentimre, which shows somewhat elevated F_ST_ relative to other Sekler populations, although the absolute values remain small. This pattern is consistent with a mild increase in genetic drift but does not amount to strong differentiation at the allele-frequency level.

TreeMix provides a complementary perspective on these subtle deviations. The inferred tree does not include any migration edges affecting Sekler groups, and all Seklers cluster within the broader Hungarian branch. However, the terminal branch leading to Nyárádszentimre is slightly elongated compared to most other Sekler populations, indicating a modest accumulation of drift over time. Deményháza, in turn, shows only minimal branch elongation and remains close to the bulk of the Sekler–Hungarian cluster. Taken together, the allele frequency–based analyses indicate that the Seklers are genetically very similar to Hungarians, with Nyárádszentimre showing a small but detectable shift, and Deményháza remaining essentially indistinguishable at this resolution.

Haplotype-based analyses, however, reveal a more pronounced and detailed internal structure. IBD sharing is elevated in both Deményháza and Nyárádszentimre relative to the other Sekler groups, whereas the remaining communities are virtually indistinguishable from the Hungarian reference population with respect to effective population size and the extent of recent shared ancestry. Elevated within-group IBD is a well-established indicator of demographic isolation, endogamy, or long-term small effective population size, and the values observed in these two villages are compatible with such scenarios. Thus, while the broader Sekler population aligns with a genetically homogeneous pattern seen across Hungarian-speaking groups, Deményháza and Nyárádszentimre show clear indications of increased local relatedness.

The IBD-based multidimensional scaling (IBD-MDS) analysis further clarifies how these two villages differ and highlights that their deviations do not follow the same direction. In the IBD-MDS space, most Sekler groups and the Hungarian reference samples form a single large cluster. Nyárádszentimre is positioned toward the upper central part of the plot, whereas Deményháza forms a separate cluster in the lower-right corner, clearly displaced from both the main Sekler–Hungarian grouping and from Nyárádszentimre. This opposite positioning shows that the two villages are not converging on a single alternative ancestry source; instead, each follows its own trajectory in haplotype space. Combined with the TreeMix results—where Nyárádszentimre has a relatively longer terminal branch but Deményháza does not—this pattern suggests that the two communities have experienced isolation and drift on different temporal scales: Nyárádszentimre shows stronger cumulative drift at the allele-frequency level, whereas Deményháza appears to have undergone more recent isolation that primarily affects haplotype sharing.

The HBD analysis provides additional context but does so in a conservative way. All Sekler groups, including Deményháza and Nyárádszentimre, fall within the typical European range with respect to total autozygosity and the distribution of homozygous segments, and none of them emerges as an extreme outlier. Both Deményháza and Nyárádszentimre tend to lie toward the upper end of the Sekler–Hungarian distribution in some HBD metrics, consistent with somewhat smaller effective population sizes or more restricted mate choice, but their values remain well within the broader European variation. These findings indicate that isolation-related drift has been present, but not to the degree observed in strongly inbred or long-standing closed populations.

Crucially, none of the allele frequency–based analyses suggests that Deményháza or Nyárádszentimre derive ancestry from external sources distinct from other Seklers or Hungarians. There is no shift toward Caucasus, Middle Eastern, Central Asian or other non-European reference populations in PCA, no population-specific ADMIXTURE component, and no migration edge targeting these groups in TreeMix. The divergence observed in the two villages is therefore best explained by village-level demographic processes, such as limited gene flow, small effective population size, and local endogamy, rather than by the retention of an older, distinct ancestry component.

Taken together, the results highlight two main points. First, the Sekler groups as a whole do not differ genetically from other Hungarian-speaking populations at the level of allele frequencies. This is consistent with historical and cultural views that describe the Seklers as a regional community with a distinct identity, rather than a population of separate genetic origin. Second, although the Sekler groups appear genetically homogeneous at the allele-frequency level, finer differences become clear when haplotype-based methods are used. Across IBD, IBD-MDS and HBD analyses, Deményháza and Nyárádszentimre stand out in a consistent way, showing the typical signatures of stronger drift, smaller effective population size and local isolation. These effects are not visible in PCA or ADMIXTURE, which explains why they only appear at the haplotype level and not in the broader allele-frequency patterns.

Although Deményháza and Nyárádszentimre lie geographically close to one another in central Transylvania, their haplotype-based patterns suggest that each community experienced its own demographic trajectory. Such differences are not unexpected at the village level: even neighboring settlements can develop distinct marriage networks, levels of endogamy, or historical population continuity, which over several generations are sufficient to produce the elevated drift signals observed here. Importantly, these local effects occur on top of a shared regional genetic background, and do not indicate deeper ancestry differences between the two villages or between them and other Sekler communities.

The combination of broad genetic homogeneity and fine-scale internal divergence is important for both methodological and historical reasons. Methodologically, it illustrates why allele frequency–based approaches alone may be insufficient to characterize small or structured populations: subtle village-level demographic histories can remain invisible in PCA or ADMIXTURE, but become apparent when haplotype sharing is examined. Biologically and historically, the results remind us that even within a culturally cohesive group such as the Seklers, demographic trajectories can diverge at the level of individual villages, reflecting local social structure, marriage patterns, geographic isolation and population size rather than major external inputs.

Finally, it is important to consider the potential impact of sample size on the population genetic inferences presented in this study. Two Sekler villages, Deményháza (*n* = 5) and Nyárádszentimre (*n* = 6), represent the smallest sample sizes among the investigated communities. Small sample sizes are known to increase variance in allele-frequency–based statistics and can lead to unstable or biased estimates of population differentiation when based solely on frequency information.

In the present study, however, the primary signals distinguishing these two settlements emerge from haplotype-based analyses, including IBD sharing, IBD-based multidimensional scaling, and HBD patterns. These approaches are specifically designed to capture recent shared ancestry and local demographic isolation and are generally more robust to moderate reductions in sample size than allele-frequency-based methods. Importantly, Deményháza and Nyárádszentimre show consistent and concordant signatures across multiple independent haplotype-based analyses, whereas no comparable deviations are observed for other villages with larger sample sizes.

Nevertheless, we note that the small number of sampled individuals from these two communities should be taken into account when interpreting village-level differences. Importantly, the observed patterns are consistent across multiple independent haplotype-based analyses, supporting their interpretation as genuine signals of local demographic structure rather than stochastic effects. While the present results provide clear evidence for increased drift and reduced effective population size in these settlements, future studies with larger sample sizes and sequence-level data may further refine quantitative estimates of these parameters.

Overall, our findings present a coherent demographic picture: the Seklers form a genetically homogeneous, regionally defined population within the broader Hungarian-speaking communities of Transylvania. Within this general homogeneity, however, there is still room for local microstructure that becomes visible only when methods capturing recent shared ancestry are applied. Deményháza and Nyárádszentimre illustrate this clearly, showing that individual villages can follow distinct demographic paths despite belonging to the same historical and linguistic community. These observations highlight that even culturally unified populations may contain fine-scale internal complexity, reflecting local demographic histories.

## 5. Conclusions

The present study supports the view that the Seklers constitute a genetically coherent and recognizable population within the Hungarian-speaking communities of Transylvania. Across a wide geographic range, individuals from different Sekler settlements share a common genetic background that is clearly identifiable and internally consistent, reflecting long-standing historical continuity rather than heterogeneous origins.

At the same time, our results demonstrate that genetic homogeneity at the regional level does not rule out the emergence of fine-scale internal structure. The microstructural patterns observed in a small number of villages do not reflect distinct ancestry components, but rather local demographic histories, such as population size, endogamy and degrees of isolation. In these communities, genetic features that are broadly characteristic of the Sekler population appear in amplified form, shaped by village-level social and demographic processes rather than by external gene flow.

These findings illustrate how culturally cohesive populations can remain genetically unified over large areas while still developing subtle internal variation at the level of individual settlements. More broadly, the study highlights the importance of considering demographic scale when interpreting population structure: what appears homogeneous at one level may reveal meaningful complexity at another. In this sense, the genetic landscape of the Seklers reflects not fragmentation, but continuity shaped by local history.

## Figures and Tables

**Figure 1 genes-17-00030-f001:**
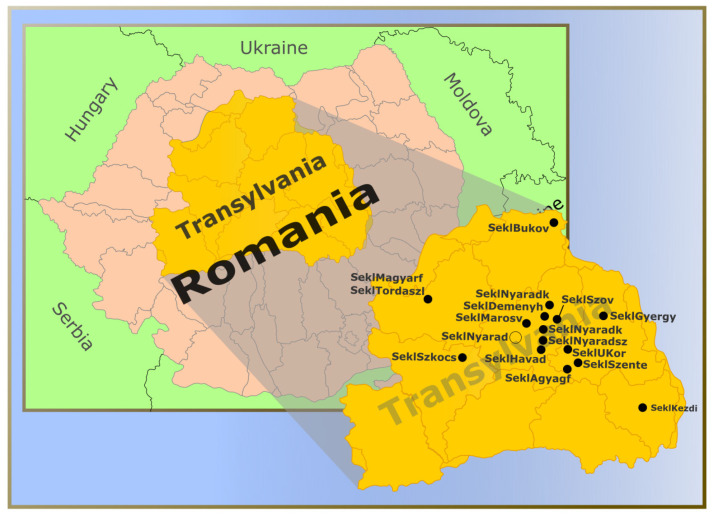
Geographic locations of the Sekler villages and the Nyárádmente region included in this study. Abbreviations used in the figure are defined as follows: SeklMagyarf—Magyarfenes; SeklTordaszl—Tordaszentlászló; SeklSzkocs—Székelykocsárd; SeklDemenyh—Deményháza; SeklMarosv—Marosvásárhely; SeklNyaradk—Nyárádköszvényes; SeklNyaradsz—Nyárádszentimre; SeklHavad—Havad; SeklNyarad—Nyárádmente; SeklSzov—Szováta; SeklBukov—Bukovina region; SeklAgyagf—Agyagfalva; SeklGyergy—Gyergyószentmiklós; SeklSzente—Szentegyháza; SeklSzudv—Székelyudvarhely; SeklUKor—Unitarians from Korond; SeklKezdi—Kézdivásárhely.

**Figure 2 genes-17-00030-f002:**
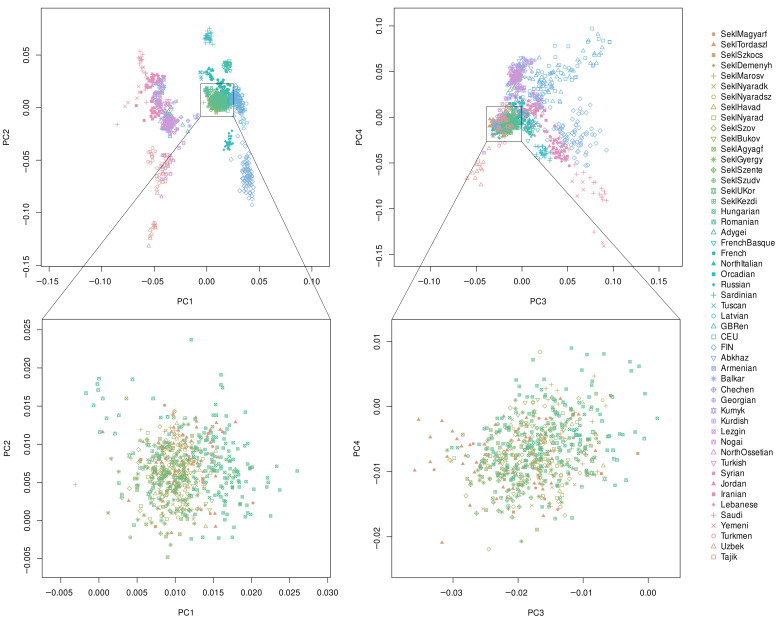
PCA analysis of the Sekler, Hungarian, Romanian, Latvian, Caucasus, Middle Eastern, Central Asian, HGDP-CEPH European, and 1 KGP European reference populations plotted along principal components 1–2 and 3–4. The bottom panels provide an enlarged view that highlights the relative positions of the Sekler, Hungarian, and Romanian samples. Eigenvalues of principal components 1–4 were 7.507, 4.152, 2.652, and 2.467, respectively. Each symbol corresponds to an individual. Appendix A presents additional PCA component plots across multiple principal component pairs, including the full set of reference populations, and Appendix A displays the scree plot of eigenvalues calculated by SmartPCA.

**Figure 3 genes-17-00030-f003:**
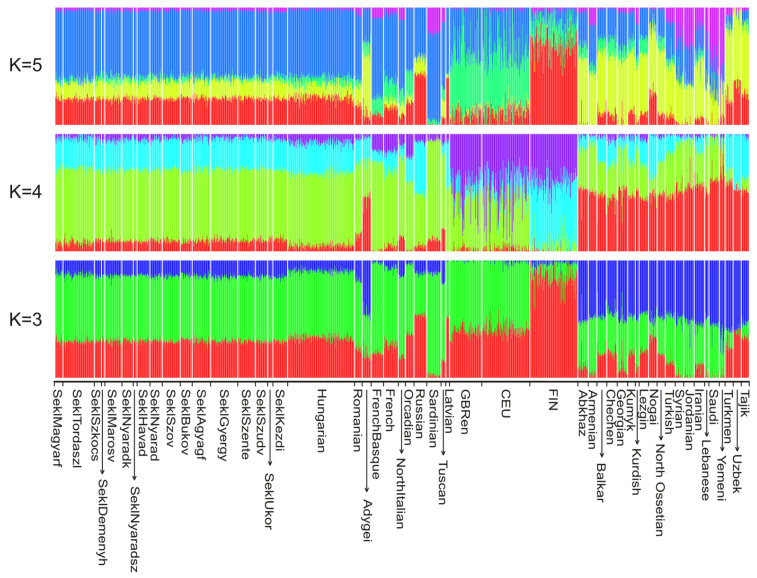
ADMIXTURE analysis results of the Sekler, Hungarian, Romanian, Latvian, Caucasus, Middle East, Central Asian, HGDP-CEPH European and the European 1 KGP reference populations with K = 3–5. The lowest cross-validation error was observed with three clusters. Cross-validation error values were 0.65927, 0.65960, and 0.65971 for K = 3, K = 4, and K = 5, respectively. Each column represents an individual, and each group of columns corresponds to a population. Appendix A shows the ADMIXTURE analysis results for K = 2 to K = 10.

**Figure 4 genes-17-00030-f004:**
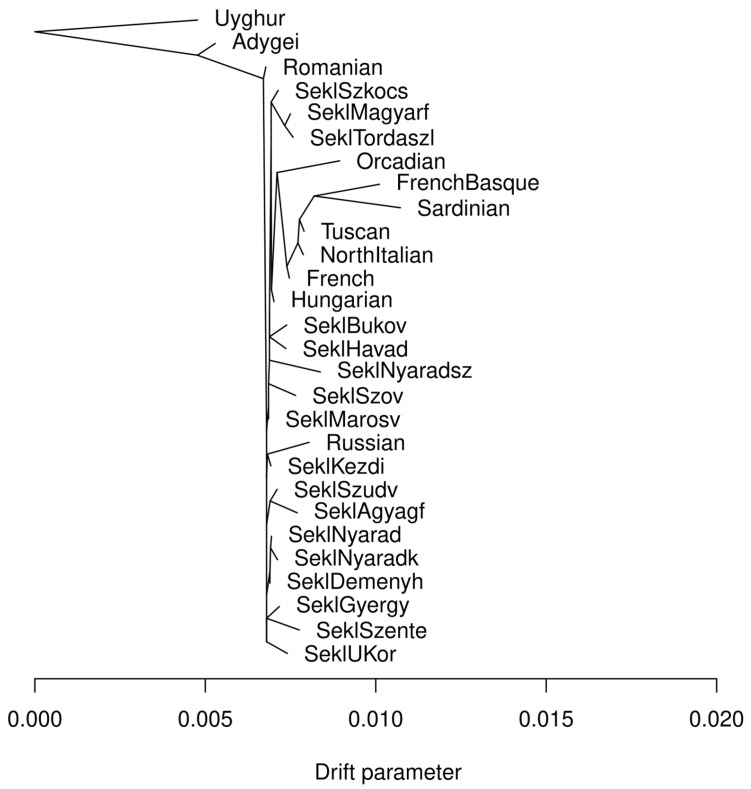
Results of the TreeMix analysis, showing the maximum likelihood tree, including Sekler, Hungarian, Romanian, HGDP-CEPH European populations and Uyghur samples served as the outgroup. Appendix A presents the residual fit from the TreeMix run.

**Figure 5 genes-17-00030-f005:**
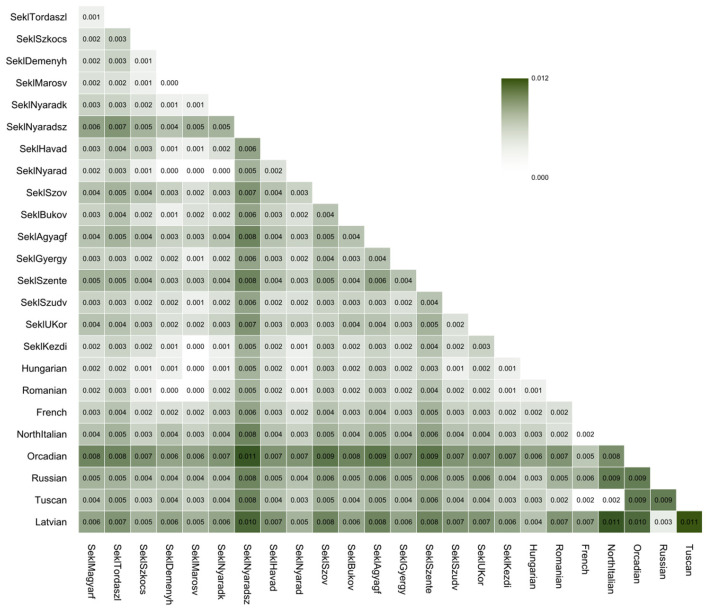
The F_ST_ matrix, calculated with SMARTPCA, illustrates the average pairwise allele frequency differentiation among the analyzed Sekler, Hungarian, Romanian, Latvian, Caucasus, Middle Eastern, Central Asian, HGDP-CEPH European, and European 1 KGP reference populations. Current figure focuses only Sekler, Hungarian, Romanian, Latvian and HGDP-CEPH European populations. Appendix A displays the F_ST_ results for all populations, and Appendix A shows the corresponding standard errors.

**Figure 6 genes-17-00030-f006:**
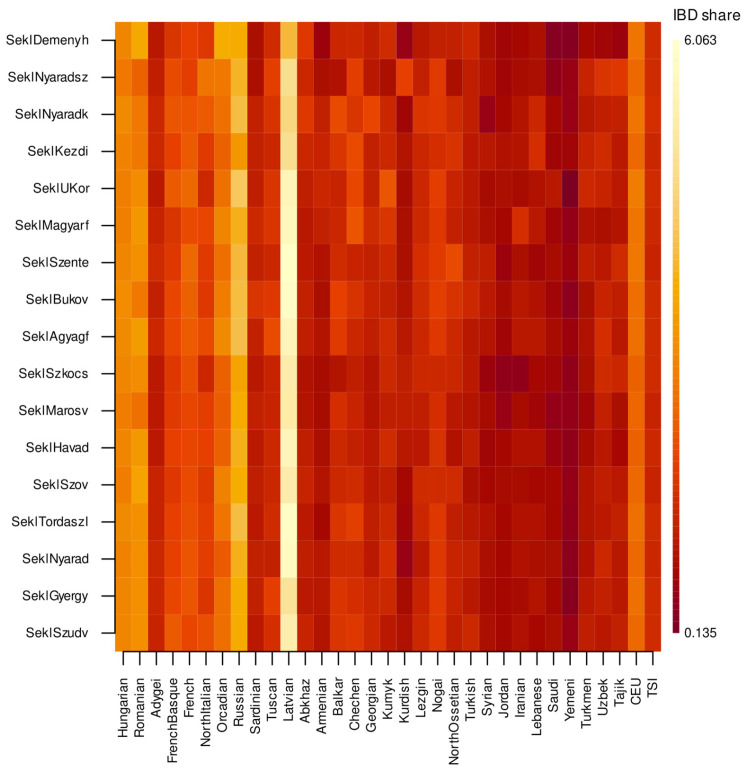
IBD segment analysis of the investigated Seklers and Hungarians, supplemented with Romanian, Latvian, Caucasus, Middle East and Central Asian samples from the Estonian Biocentre, the HGDP-CEPH European samples and HapMap Phase 3 CEU and TSI samples as reference populations.

**Figure 7 genes-17-00030-f007:**
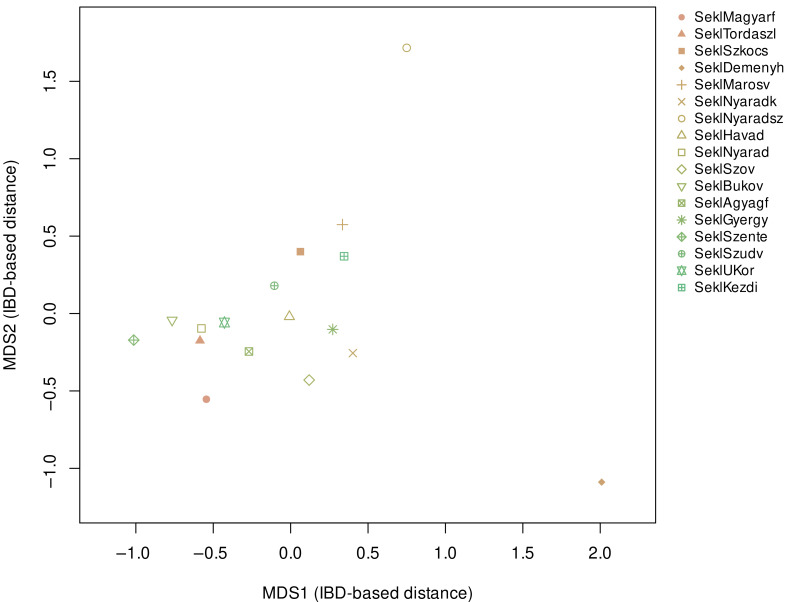
MDS plot of Sekler IBD-sharing profiles. Calculations are based on previous IBD-share results.

**Figure 8 genes-17-00030-f008:**
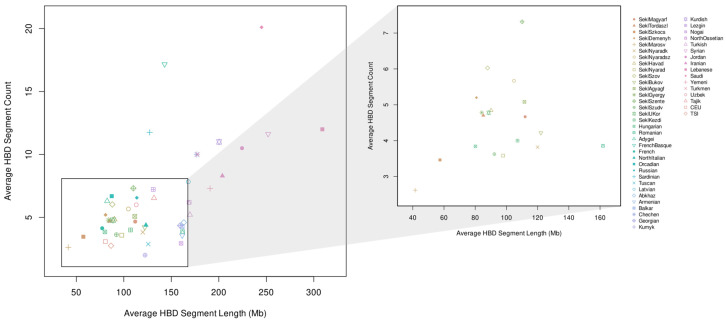
Genome-wide average autozygosity of the investigated Seklers and Hungarians, supplemented with Romanian, Latvian, Caucasus, Middle East and Central Asian samples from the Estonian Biocentre, the HGDP-CEPH European samples and HapMap Phase 3 CEU and TSI samples as reference populations. The right panel provides an enlarged view highlighting only the Sekler, Hungarian and Romanian groups to facilitate visualization of fine-scale differences.

## Data Availability

All data generated or analyzed during this study are included in this published article and its Appendix A. Some of the datasets are available in public online repositories. The HGDP data are available directly from the homepage of Rosenberg lab at the Stanford University (https://rosenberglab.stanford.edu/hgdpsnpDownload.html, accessed on 23 December 2025), while the 1 KGP (https://www.internationalgenome.org/category/vcf/, accessed on 23 December 2025) data are available directly from their ftp server (https://ftp.1000genomes.ebi.ac.uk/vol1/ftp/release/20130502/, accessed on 23 December 2025). The HapMap Phase 3 dataset can be downloaded from the International HapMap Project repository hosted by the Wellcome Sanger Institute (https://www.sanger.ac.uk/data/hapmap-3/, accessed on 23 December 2025). Populations of the Estonian Biocentre can be downloaded from their repository (https://evolbio.ut.ee/, accessed on 23 December 2025). The Sekler and Hungarian datasets, according to the Hungarian Human Genetics Act 2008/XXI, cannot be uploaded to a public online database, but can be obtained upon reasonable request via e-mail from the corresponding authors.

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
