# Peer review of "Uncovering the Genetic Structure of the Sekler Population in Transylvania Through Genome-Wide Autosomal Data"

_genes, 2025, doi:10.3390/genes17010030_

Round 1
Reviewer 1 Report
Comments and Suggestions for Authors
Methods used in this manuscript are fine. However, I would like to recommend the authors additional analysis. That is analysis with f statistics, especially f3. As you may know, the methods estimate the drift difference between populations. As you described two village population showed the difference in genetic drift, you could show directly this difference by the f3 statistics.
And one more recommendation, you had better to compare the nucleotide diversity (pi value) and homozygosity at the nucleotide level of the two village populations and show the difference of the effective size and mating system by the extent of the value of the diversity.
Furthermore, the sample size of the two village populations are 5 and 6, the two most smallest sample number among 17 villages. For other villages, the sample numbers are 13 to 64. I could not find any discussions about the effect of sample numbers on the statistics used in this manuscript. I recommend the authors to discuss the effect of sample number on the statistics.
Reviewer 2 Report
Comments and Suggestions for Authors
The manuscript presents the genetic structure of the Sekler population from Transylvania using genome‑wide autosomal SNPs data. After quality control and exclusion of non-eligible samples, a final dataset of 595 individuals from 17 villages was analyzed and merged with several European and Eurasian reference panels.
Allele frequency analyses show that all Sekler groups form a single cluster that widely overlaps with the Hungarian reference population, indicating shared ancestry and no substantial admixture from non‑European sources. Haplotype based analyses reveal localized signals of genetic drift and isolation in two villages, Deményháza and Nyárádszentimre.
Overall, the results suggest that the Sekler population represents a regionally distinct but genetically homogeneous subgroup within the Hungarian speaking population gene pool, while localized demographic processes have produced a detectable substructure in a few isolated villages.
Comments
The manuscript is well structured and meets the standards expected for a population genetic study. The analytical framework is appropriate for the study focus. The quality of the English language is good and suitable for publication.
However, there are some minor issues to address:
● In the “Materials and methods” section, the description of biological sample collection, who performed the sampling, the DNA extraction procedures and the DNA quantification step is missing. These aspects areimportant to ensure analytical reproducibility and should be described in more detail. In addition, the laboratory where the genetic analyses were performed is not reported (see line 122). This information should be clearly stated. ● In the text, the company's details are not reported. When mentioning for the first time a commercial system, instrument, software or anything trademarked it is necessary to properly cite the manufacturer and its headquarters’ information. An example could be: “Applied Biosystems™ 3500 Genetic Analyzer” (Thermo Fisher Scientific, Waltham, MA, USA). I recommend fixing this issue. ● Although the “Discussion” section is deeply described, a dedicated Conclusions section is missing. Adding a short conclusion section summarizing the main findings would improve clarity and align the manuscript with standard publication format.Overall, the study addresses a relevant topic for the field, the sample size is adequate and no major interpretative or methodological issues were identified. I therefore suggest a minor revision.
